# 3S-Attack: Spatial, Spectral and Semantic Invisible Backdoor Attack Against DNN Models

## Abstract

Backdoor attacks implant hidden behaviors into models by poisoning training data or modifying the model directly. These attacks aim to maintain high accuracy on benign inputs while causing misclassification when a specific trigger is present. While existing studies have explored stealthy triggers in spatial and spectral domains, few incorporate the semantic domain. In this paper, we propose 3S-attack, a novel backdoor attack which is stealthy across the spatial, spectral, and semantic domains. The key idea is to exploit the semantic features of benign samples as triggers, using Gradient-weighted Class Activation Mapping (Grad-CAM) and a preliminary model for extraction. Then we embedded the trigger in the spectral domain, followed by pixel-level restrictions in the spatial domain. This process minimizes the distance between poisoned and benign samples, making the attack harder to detect by existing defenses and human inspection. And it exposes a vulnerability at the intersection of robustness and semantic interpretability, revealing that models can be manipulated to act in semantically consistent yet malicious ways. Extensive experiments on various datasets, along with theoretical analysis, demonstrate the stealthiness of 3S-attack and highlight the need for stronger defenses to ensure AI security.

## 1 Introduction

With the rapid integration of artificial intelligence (AI) into diverse sectors such as finance, healthcare, and daily life, concerns about the security and trustworthiness of AI systems are intensifying. An increasing number of studies have revealed the vulnerabilities of AI models, raising concerns about their reliability in real-world applications. Among them, backdoor attacks have drawn significant attention due to their stealthy nature and minimal deployment cost Gu et al. (2019). In a typical backdoor attack, an adversary poisons a small subset of the training data by injecting inputs containing a specific trigger and labeling them with the target class. Once trained, the model performs well on benign inputs but misclassifies any input with the trigger into the attacker-specified class. Notably, modifying as little as 1% of the training data is sufficient to embed a backdoor Gu et al. (2019), and the entanglement of backdoor functionality with normal neurons further complicates detection and removal. Consequently, defending against such attacks forms a pressing challenge.

In neural networks, spatial domain refers to the arrangement of pixels in an image, spectral domain focuses on frequency components of samples (e.g., via Fourier transforms), and semantic domain captures latent features of sample generated by pre-defined metrics or the model. Over the years, various defense strategies have emerged, targeting different domains: spatial Wang et al. (2019), spectral Zeng et al. (2021), and semantic Liu et al. (2018a) characteristics. In response, attackers have developed more covert strategies to optimize the stealthiness of the trigger and backdoor attack across specific domains Nguyen & Tran (2021); Feng et al. (2022), seeking to evade these defenses and human inspections. However, existing attacks have never considered stealthiness in all three domains simultaneously. And existing semantic-aware attacks either require access to the training process Zhong et al. (2022); Cheng et al. (2021), or fail to achieve stealth across multiple domains simultaneously.

To address these limitations, we propose 3S-Attack, a novel backdoor attack that achieves stealthiness across three complementary domains: spatial, spectral, and semantic. Our method requires no access to the training pipeline. Instead, it operates solely through data poisoning. Leveraging

Grad-CAM Selvaraju et al. (2017), we extract the semantic features of benign class samples and embed them into the poisoned images. We then restrict pixel-level perturbations to preserve visual indistinguishability. The resulting poisoned samples remain nearly identical to clean ones in appearance, frequency characteristics, and high-level features, effectively evading both human perception and state-of-the-art defense techniques.

As illustrated in Figure 1, 3S-Attack introduces less perturbation to both spatial space and spectral space compared to widely adopted backdoor methods (as can be seen by less light-up points in the images), while achieving strong attack success rate.

The main contributions of this work are as follows:

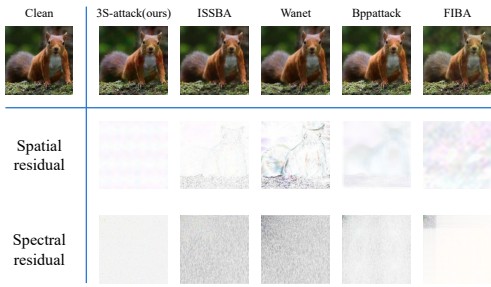

1. We propose 3S-Attack, the first backdoor attack to *simultaneously* achieve stealthiness in spatial, spectral, and semantic domains.

2. 3S-Attack is also the first semantic-domain stealthy backdoor attack that operates purely through poisoned samples, without requiring access to the model training process.

3. Extensive experiments and theoretical analysis demonstrate the superior stealth compared to prior state-of-the-art attacks and defense-resistance capabilities of our proposed method.

Figure 1: Comparison of proposed 3S-attack with other SOTA backdoor attacks in spatial and spectral perspective. The residual is the difference between benign and poisoned samples, and color reversed for better demonstration.

## 2 BACKGROUND AND RELATED WORK

Research on backdoor attacks can be divided into two categories: attack schemes and corresponding defense methods.

### 2.1 EXISTING BACKDOOR ATTACKS

A typical backdoor attack defines a trigger and target class, then selects samples from non-target classes, embeds the trigger, and relabels them as the target class. These poisoned samples are added to the training set, causing the model to learn a hidden association between the trigger and the target class during training. In 2017, Gu et al. Gu et al. (2019) first proposed BadNets, defining the concept of backdoor attacks targeting DNN models and revealing their potential risks. Since then, a plethora of research papers on backdoor attacks have emerged. Currently, research on backdoor attacks can be categorized into two stages: visible backdoor attacks and invisible backdoor attacks.

At the stage of visible backdoor attack, attackers primarily focus on enhancing the reliability and attack success rate (ASR) of the attack Chen et al. (2017); Barni et al. (2019); Lovisotto et al. (2020); Liu et al. (2021), paying less attention to whether the trigger is conspicuous, i.e., whether it can be detected by human observation or defense methods.

Meanwhile, after the concept of backdoor attacks was introduced, numerous researchers began developing defense methods. Consequently, as understanding of backdoor attacks deepened, human-recognizable triggers were gradually abandoned due to their susceptibility to detection. During this stage, researchers not only ensured the effectiveness of the attack but also emphasized improving its stealthiness. This includes invisibility to defense methods Xue et al. (2020), i.e., bypassing various defenses, and invisibility to humans Zhong et al. (2020); Zou et al. (2018); Wang et al. (2022), ensuring the poisoned samples appear normal and coherent to the human eye. Subsequently, researchers found that adding backdoors to the frequency domain features of samples can make the poisoned samples inherently covert in the spatial domain. Therefore, some researchers have attempted to implement backdoor attacks from the frequency perspective Yu et al. (2023); Gao et al. (2024); Zeng et al. (2021).

In addition to adding poisoned samples to training datasets, researchers have also explored implanting backdoors by directly modifying models Tang et al. (2020) or the training environment Doan et al. (2021). If the attacker is a provider of Machine Learning as a Service (MLaaS), they can access both the training dataset and the training process, enabling more efficient and stealthy backdoor attacks Liu et al. (2018b); Zhong et al. (2022). Beyond image classification tasks, recent work has extended backdoor attacks to models performing other tasks, including backdoor attacks on transfer learning Yao et al. (2019), federated learning Bagdasaryan et al. (2020), self-supervised Saha et al. (2022) and semi-supervised learning Yan et al. (2021), as well as models for voice recognition Shi et al. (2022) and natural language processing Cheng et al. (2025).

## 2.2 Existing Backdoor Defenses

The existing backdoor defense methods can be categorized into three types based on their focus: spatial domain-based, spectral domain-based, and semantic domain-based backdoor defenses.

**Spatial Domain** In image classification tasks, the spatial domain refers to the arrangement of pixels within each sample image. Backdoor defense methods that analyze from the spatial perspective attempt to detect backdoors directly without applying any transformations to the samples or the model. Their techniques often involve reverse engineering the trigger Wang et al. (2019), overlapping Gao et al. (2019), analyzing model attentions Selvaraju et al. (2017); Chou et al. (2020), and blocking Doan et al. (2020).

**Spectral Domain** Spectral-based backdoor defense methods involve transforming image samples from the spatial domain to the frequency domain using techniques such as FFT (Fast Fourier Transform) or DCT (Discrete Cosine Transform). After transformation, these methods identify triggers and backdoors by detecting abnormal changes in frequencies and amplitudes Zeng et al. (2021); Fu et al. (2021) such as abnormal clustering Hammoud et al. (2023), caused by trigger insertion. They leverage frequency-domain features to achieve efficient and robust real-time detection.

**Semantic Domain** The semantic domain refers to any space that can represent or maximize the features of a sample. This domain can include not only manually defined spaces but also those automatically discovered by the model during training. For instance, to classify samples more efficiently, the model often assigns one or more neurons to specific features. In this case, the set of neurons activated by a sample can be regarded as its representation in the model's abstract semantic domain. Based on analyzing these activations, poisoned samples can be identified Chen et al. (2018); Liu et al. (2019); Tran et al. (2018), and backdoor model can also be fine-tuned to remove the backdoor Liu et al. (2018a); Li et al. (2021a).

## 3 3S-attack

In this paper, we focuses on designing a backdoor trigger that is stealthy in the spatial, spectral, and semantic domains. We name our attack *3S-attack*, as it satisfies all the above requirements. Moreover, existing backdoor attacks that achieve semantic-domain stealth typically require control over the training process or access to model parameters Zhong et al. (2022), which is impractical in many real-world scenarios. To the best of our knowledge, this is also the first attack to achieve semantic stealth without access to the model or training pipeline. This advancement significantly broadens the feasibility of advanced backdoor attacks and poses a serious challenge to existing defense strategies.

### 3.1 Threat Model

In this work, we follow the most common assumptions adopted in previous studies Gu et al. (2019); Nguyen & Tran (2021); Wang et al. (2022); Feng et al. (2022); Chen et al. (2017); Lovisotto et al. (2020); Xue et al. (2020); Li et al. (2021b); Zou et al. (2018); Yu et al. (2023); Gao et al. (2024); Doan et al. (2021); Liu et al. (2018b); Yan et al. (2021); Saha et al. (2022).

**Attacker's Capability** The attacker can inject or alter a certain number of samples in the training dataset. For example, the attacker may generate poisoned samples and publish them online, waiting

for victims to collect them as part of their training dataset; or the attacker may be a third-party data collection or labeling service provider who has more control over the victim's dataset. However, we do not assume that the attacker has access to the model itself, such as the training process, model parameters, or loss function. This is because very few individuals or parties have access to a specific victim's model, and attacks conducted by such parties are highly traceable. Therefore, in this paper, we assume that the attacker has access to the training dataset but not to the model training process.

**Attacker's Goal** The attacker's goal is to successfully implant a backdoor into the target model via data poisoning. Specifically, the attacker designs a trigger, generates multiple poisoned samples using it and change their label to the target class, and relies on the victim to train a model with these samples. The backdoor attack should fulfill the following characteristics: feasibility (remains inactive on benign samples but causes misclassification to a target class when triggered), stealthiness (undetectable through human inspection across various domains), and defense resistance (resistant to defenses from different perspectives).

## 3.2 ATTACK METHOD INTUITION

The major challenge in designing a stealthy backdoor attack lies in making the trigger invisible in the semantic domain, which is an abstract space autonomously learned by the model and exhibits strong black-box characteristics. It is impossible to predict the shape of this space before training begins, let alone describe it accurately.

To address this challenge, the proposed 3S-attack adopts a strategy of *fighting magic with magic*. Theoretically, a fully trained model should focus on parts of an input image that best reflect the features associated with its label Selvaraju et al. (2017). For instance, if an image is labeled as a *cat*, the model should focus on the parts that reveal the presence of a cat (e.g., the cat's body), while ignoring irrelevant parts (e.g., the background). Hence, models trained on similar datasets are expected to focus on roughly the same regions when classifying the same sample. Further detailed theories and experiments can be found in the appendix A.3.

Following this insight, the 3S-attack attempts to indirectly characterize the semantic domain and bypass the difficulty of describing it, by leveraging a preliminary model to predict the semantic domain of the target model. Specifically, the attacker first trains a clean model. *No specific requirements are imposed on this clean model*, as long as it achieves acceptable classification accuracy. Then apply Grad-CAM to compute saliency map for given samples. By analyzing the saliency map, the attacker identifies the parts of the image that are most important to the model. This information is then utilized to construct the trigger for the backdoor attack. The detailed procedures for trigger generation and injection are introduced below.

## 3.3 STEP 1: TRIGGER EXTRACTION

The attacker firstly trains a preliminary model using a clean dataset. This model need not achieve optimal accuracy—only an acceptable performance level, and its structure can be different with the target model. The attacker then selects a *target class* and chooses one or more *trigger samples* from this class to generate the trigger. As shown in Figure 2, the attacker uses Grad-CAM to extract the regions the model relies on most when classifying the trigger samples (saliency maps). These saliency maps are

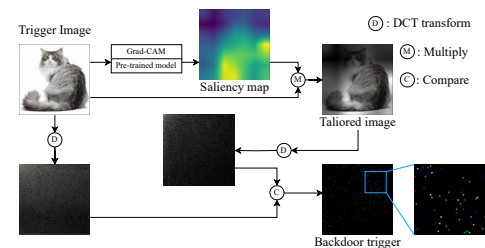

Figure 2: Pipeline for extracting a trigger from a benign sample in target class.

multiplied with the corresponding samples to produce *tailored samples*. Both the original trigger samples and the tailored samples are then transformed using the Discrete Cosine Transform Ahmed et al. (1974)(DCT). The attacker compares the magnitude of each frequency component in the resulting spectrograms. Frequencies with magnitude differences below a certain threshold (named *Frequency Selection Threshold*) are considered the key features that the model uses for prediction. These frequencies and the corresponding magnitudes are chosen as the trigger. The further explain

on why stable DCT components represent the semantic feature is in Appendix A.4 and analysis of time expenditure for each step is in Appendix A.5.

## 3.4 STEP 2: POISON SAMPLES GENERATION

The next step is to inject the trigger into samples to generate poisoned samples. As shown in Figure 3, for a target sample, the attacker first applies DCT to obtain its spectral map. Then, the magnitudes of the trigger-identified frequencies in the target sample are adjusted towards the corresponding values in the trigger, based on a predefined extent of *Poison Distance Ratio*. After this adjustment, inverse DCT is applied to convert the sample back into the spatial domain. The pseudocode of 3S-attack is provided in Algorithm 1 in Appendix.

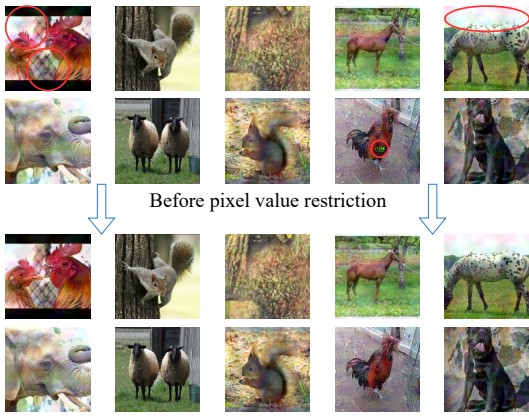

Figure 3: Process of embedding the trigger into benign samples to generate poisoned samples.

## 3.5 STEP 3: PIXEL CHANGE RESTRICTION

However, preliminary experiments indicate that directly adding triggers in the spectral domain can result in unnatural artifacts in the spatial domain (see the upper half of Figure 4). Therefore, it is necessary to constrain pixel variations. Specifically, after inverse transformation, the modified sample is compared with the original in terms of pixel values. If the change in any pixel exceeds *Pixel Change Restriction Threshold*, the change is limited to that threshold. The same rule applies when pixel values exceed the data boundaries (e.g., 0–255 for uint8, or 0–1 for float data). Note that the pixel value change restriction step does not always take effect, as in most cases the pixel changes caused by trigger injection do not exceed the pixel change threshold. In other words, the pixel restriction serves merely as a safeguard in case the pixel changes become too large.

Figure 4: Pixel value change restriction on poisoned samples. Note that the red circles in the figure are solely used to highlight the unnatural artifacts in the samples; the circles themselves are not part of the poisoned samples.

## 4 EXPERIMENTS

### 4.1 EXPERIMENTAL SETUP

**Environment** All experiments were conducted on a server equipped with NVIDIA A100 Tensor Core GPUs and Intel® Xeon® Platinum 8570 CPUs, running Red Hat Enterprise Linux 8.10. All experiments were performed using Python 3.11.0 and PyTorch 2.5.1+cu118. We used the Adam optimizer with a learning rate of 0.001, a batch size of 128, and trained the models for 50 epochs.

**Datasets** Backdoor attacks against DNNs have mainly focused on models for image classification tasks. Therefore, we selected datasets that are representative in the image classification field to demonstrate the generalizability of the 3S-attack across various scenarios.

We strategically selected MNIST LeCun et al. (1998), GTSRB Stallkamp et al. (2011), CIFAR-10 Krizhevsky et al. (2009), CIFAR-100 Krizhevsky et al. (2009), Animal-10 Song et al. (2022), and Imagenet Deng et al. (2009) to comprehensively evaluate our attack's feasibility, stealthiness, and resistance to defenses. In Imagenet dataset, due to the limited computational resources, we have

chosen a subset of it by randomly selecting 20 classes from the entire dataset. Table 1 shows the detailed statistics of each dataset.

The selection of these datasets was based on key considerations to ensure a thorough and balanced evaluation. Their structured diversity—encompassing handwritten digits, traffic signs, objects, animals, and high-resolution images—ensures a robust evaluation of 3S-attack across varied conditions.

| Dataset | Input Size | #Train | #Test | Classes |
|---|---|---|---|---|
| MNIST | $28 \times 28 \times 1$ | 60000 | 10000 | 10 |
| GTSRB | $32 \times 32 \times 3$ | 39209 | 12630 | 43 |
| CIFAR-10 | $32 \times 32 \times 3$ | 50000 | 10000 | 10 |
| CIFAR-100 | $32 \times 32 \times 3$ | 50000 | 10000 | 100 |
| Animal-10 | $128 \times 128 \times 3$ | 23679 | 2500 | 10 |
| Imagenet | $224 \times 224 \times 3$ | 26000 | 1000 | 20 |

Table 1: Details of each dataset.

**Models** We employ different neural network architectures tailored to the complexity and characteristics of each dataset to ensure a meaningful and rigorous evaluation. For MNIST, we utilize both a custom small model and LeNet-5, as this dataset consists of low-resolution grayscale images of digits, requiring relatively simple architectures. For GTSRB and CIFAR-10, we use VGG-11 and ResNet-18 to study the impact of model depth and feature extraction strategy. For CIFAR-100, we use WideResNet (WRN), a ResNet variant with wider layers, offering greater feature representation capacity. For Animal-10, we adopt ResNet-18 due to its balance between capacity and efficiency. For Imagenet, we adopt ResNet-50 because of the complexity and high-resolution.

**Metrics** We mainly use Attack Success Rate (ASR), Peak Signal-to-Noise Ratio (PSNR), and Structural Similarity Index Measure (SSIM) to measure the effectiveness and stealthiness of 3S-attack. ASR Gu et al. (2019) measures the effectiveness of a backdoor attack by quantifying the probability that a model misclassify poisoned samples as the target class when the trigger is present. PSNR Hore & Ziou (2010) evaluates the stealthiness of a backdoor trigger in pixel level by measuring the pixel level similarity between the original and poisoned samples. SSIM Hore & Ziou (2010) assesses the perceptual similarity between the original and poisoned images in global level, considering not only pixel-wise differences but also structural information.

## 4.2 Attack Performance Evaluation

**Baseline Attack** We selected several baseline backdoor attack methods that employ different trigger and poisoned sample generation algorithms to compare with 3S-attack. Specifically, we chose the following attack method to compare with. Wanet Nguyen & Tran (2021) defines a warping field as the trigger and applies it to benign samples, it acts in the spatial domain. Bppattack Wang et al. (2022) uses quantization and dithering as the trigger mechanisms, it acts in the spatial domain. ISSBA Li et al. (2021b) trains an encoder-decoder pair initially designed for steganography to embed hidden triggers, it acts in the semantic domain. FIBA Feng et al. (2022) selects the central frequencies of a benign sample as trigger and replaces that of other samples to generate poisoned inputs, it acts in the spectral domain. BadNets Gu et al. (2019) serves as a standard baseline and is used to evaluate the effectiveness of various defenses, it acts in the spatial domain.

| | Clean | 3S-attack | | ISSBA | | Wanet | | Bppattack | | FIBA | | DUBA | |
|---|---|---|---|---|---|---|---|---|---|---|---|---|---|
| Datasets | BA | BA | ASR | BA | ASR | BA | ASR | BA | ASR | BA | ASR | BA | ASR |
| MNIST | 99.34 | 99.20 | 96.47 | 99.23 | **99.10** | 98.83 | 97.43 | - | - | 99.18 | 70.68 | 99.12 | 95.14 |
| GTSRB | 98.36 | 96.55 | 94.12 | 97.38 | 93.71 | 96.89 | **98.31** | 97.15 | 95.29 | 97.07 | 79.05 | 97.83 | 97.26 |
| CIFAR10 | 86.40 | 84.65 | 89.29 | 84.80 | 77.23 | 85.13 | **93.36** | 85.54 | 91.32 | 84.93 | 65.85 | 86.97 | 95.79 |
| CIFAR100 | 66.94 | 66.64 | 92.38 | 66.39 | 86.42 | 66.05 | **93.06** | 66.26 | 85.94 | 66.78 | 75.48 | 66.21 | 96.78 |
| Animal10 | 88.08 | 87.32 | 97.42 | 86.96 | **99.87** | 87.52 | 93.88 | 86.84 | 92.44 | 87.36 | 58.72 | 88.04 | 98.30 |
| Imagenet | 74.80 | 72.60 | 88.21 | 72.70 | 82.74 | 74.30 | 87.16 | 73.20 | 89.53 | 71.80 | 73.84 | 72.20 | **92.53** |

Table 2: BA and ASR value of different attacks in spatial domain. Note that the BA and ASR is in percentage format.

**Attack Performance** We compare the proposed 3S-attack with other baseline backdoor attacks, and Table 2 demonstrates the attack affectiveness by showing the Benign Accuracy (BA) and Attack Success Rate (ASR), while Table 3 demonstrates the attack stealthiness by showing the PSNR and SSIM value of each backdoor attack across the above mentioned datasets. During experiments, we adopted different poison rate for each dataset to achieve the best attack result. Specifically, we used

| Datasets | 3S-attack | | ISSBA | | Wanet | | Bppattack | | FIBA | | DUBA | |
|---|---|---|---|---|---|---|---|---|---|---|---|---|
| | PSNR | SSIM | PSNR | SSIM | PSNR | SSIM | PSNR | SSIM | PSNR | SSIM | PSNR | SSIM |
| MNIST | **46.01** | **0.943** | 39.22 | 0.892 | 34.13 | 0.639 | - | - | 23.93 | 0.679 | 38.19 | 0.893 |
| GTSRB | **32.78** | **0.979** | 19.03 | 0.653 | 31.22 | 0.759 | 24.61 | 0.943 | 14.62 | 0.559 | 31.58 | 0.889 |
| CIFAR10 | **35.65** | **0.969** | 23.51 | 0.852 | 29.95 | 0.773 | 20.06 | 0.923 | 15.50 | 0.710 | 31.98 | 0.918 |
| CIFAR100 | **31.68** | **0.946** | 22.79 | 0.851 | 30.69 | 0.858 | 20.12 | 0.927 | 15.87 | 0.770 | 30.02 | 0.909 |
| Animal10 | 30.83 | **0.962** | 26.60 | 0.840 | 29.59 | 0.452 | 23.28 | 0.966 | 15.69 | 0.754 | **32.69** | 0.916 |
| Imagenet | **32.82** | 0.963 | 31.39 | 0.875 | 28.13 | 0.129 | 23.51 | **0.967** | 17.69 | 0.776 | 32.33 | 0.885 |

Table 3: PSNR, and SSIM value of different attacks in spatial domain.

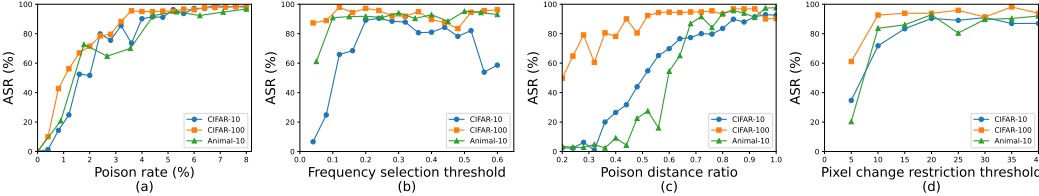

Figure 5: The effect of (a) poison rate, (b) frequency selection threshold, (c) poison distance ratio, and (d) pixel change restriction threshold on ASR, evaluated on three datasets: CIFAR-10, CIFAR-100, and Animal-10.

the poison rate of 1% for MNIST, 2% for GTSRB, 4% for CIFAR-10, CIFAR-100, and Animal-10 dataset. Specifically, the 3S-attack attains a consistently high ASR across each datasets, showing that it is achieving a acceptable attack feasibility. More importantly, it demonstrates remarkably high PSNR and SSIM scores across all datasets. These results indicate that the perturbations introduced by 3S-attack are not only effective but also imperceptible. Therefore, compared with baseline attacks which sacrifice trigger stealthiness for ASR, 3S-attack offers a better trade-off between effectiveness and imperceptibility. Besides, 3S-attack is having a constant performance across different dataset, indicating its strong generalization capability. Note that the results of BppAttack on MNIST are omitted because BppAttack is incompatible with the data characteristics of MNIST. Specifically, most pixel values in MNIST are either 0 (the minimum) or 255 (the maximum), resulting in a highly saturated dataset. When BppAttack is applied, it often produces pixel values that exceed these limits. Due to value clipping, any modifications that fall outside the valid pixel range are suppressed, rendering the inserted triggers ineffective. As a result, BppAttack consistently fails to generate valid poisoned samples on MNIST.

### 4.3 PARAMETERS

In 3S-attack, there are multiple parameters and thresholds that can affect the performance of the attack. Among them, the most important parameters are:

1. Poison rate: This parameter measures how much rate of samples in the training dataset are changed into poisoned, in order to embed the backdoor.

2. Frequency Selection Threshold: When comparing the frequency map of target and tailored images, frequencies with enough similarity are selected as part of the trigger.

3. Poison distance ratio: When injecting the trigger into samples, the amplitude of trigger frequencies in benign sample will move towards the value in trigger in this specific extent.

4. Pixel change restriction threshold: This parameter controls to what extent the tolerance is on pixel value change on poisoned samples.

We evaluate how these parameters affect 3S-attack performance. Figure 5, illustrate the effects on ASR for CIFAR-10, CIFAR-100, and Animal-10, respectively. Generally, ASR monotonically increases as the proportion of trigger components in the dataset increases, which is intuitive—higher poisoned intensity leads to higher ASR. And with the increase of number of poisoned samples (sub-figure a), frequency selection threshold (sub-figure b), poison distance ratio (sub-figure c), and pixel change restriction threshold (sub-figure d), the proportion of trigger components in the dataset increases.

As illustrated in the Fig 5, Table 2, and Table 3, the 3S-attack exhibits strong stealthiness and robustness. Even when the proportion of trigger components in the dataset is low corresponding to conservative parameter settings, it consistently achieves a satisfactory ASR. Furthermore, it maintains a relatively high ASR across a wide range of parameter variations. Specifically, when frequency selection threshold $\in [0.15, 0.5]$, poison distance ratio $\in [0.7, 1]$, and pixel change restriction threshold $\in [0.1, ]$, the 3S-attack stay effective while not decreasing the BA of the victim model. These results suggest that effective attack performance can be attained with high probability, even in the absence of detailed knowledge about the target model or dataset. This highlights the practicality of the 3S-attack, as its parameters can be configured based on general intuition or prior experience rather than precise model-specific tuning, which enhanced the robustness of 3S-attack.

## 4.4 ABLATION STUDY

We theoretically and experimentally assess the 3S-attack's performance when each key component is removed.

**Grad-CAM vs Random Frequency Pick**  If frequencies in the DCT map are selected randomly, rather than according to a substitute model and the Grad-CAM method, as introduced in Section 3.3, the trigger can still generate poisoned samples and embed a backdoor. However, neither the trigger nor the poisoned samples will contain any features related to the target class. Consequently, they cannot achieve invisibility in the semantic domain. The same theory holds when a random area is picked instead of calculate the saliency map to select the model focusing area. The results are shown in Table 4, where Grad-CAM guided frequency selection strategy and random frequency selection strategy achieved similar attack effectiveness and stealthiness. But the random frequency selection strategy resulted in a much higher $MMD^2$ score, indicating that without Grad-CAM, the poison samples generated rarely share semantic features with benign samples in target class, which will diminish the invisibility of backdoor attack in semantic domain.

| Strategy | BA | ASR | PSNR | SSIM | $MMD^2$ score |
|---|---|---|---|---|---|
| Grad-CAM | 85.03 | 89.05 | 35.53 | 0.963 | 0.5984 |
| Random Frequency Pick | 84.83 | 88.75 | 35.16 | 0.954 | 0.9614 |

Table 4: Comparison of attack performance between Grad-CAM guided frequency selection, and random frequency selection.Note that the BA and ASR is in percentage format.

**Trigger Shifting**  Because the DCT is a linear transform, directly adding a universal trigger to each sample is equivalent to attaching a kind of same pattern to every sample in the spatial domain, which causes the trigger not sample-specific. Moreover, most frequencies in the DCT maps of natural images have zero amplitude. Replacing these amplitudes with those of the trigger has a similar effect. To ensure sample specificity, additional nonlinearity must be introduced; this makes shifting each sample toward the trigger amplitudes (as described in Section 3.4) necessary.

| Strategy | BA | ASR | PSNR | SSIM |
|---|---|---|---|---|
| Pixel Restriction On | 85.03 | 89.05 | 35.53 | 0.963 |
| Pixel Restriction Off | 85.65 | 90.86 | 34.18 | 0.951 |

Table 5: Comparation between if pixel restriction step is engaged or not.Note that the BA and ASR is in percentage format.

**Pixel Value Restriction Switch on/off**  When the other parameters are set to reasonable values, the pixel value restriction introduced in Section 3.5 has only a marginal influence on overall performance. As previously introduced, its primary function is to serve as a safeguard that prevents excessively large pixel deviations after the inverse transformation. Consequently, as shown in Table 5, disabling this restriction slightly increases the ASR, since none of the frequency-domain trigger components are attenuated by clipping. However, the absence of this control allows occasional large pixel fluctuations to persist, leading to a minor decrease in averaged PSNR and SSIM

value. This reflects the trade-off between maintaining trigger fidelity and constraining unintended visual perturbations.

## 4.5 DEFENSE RESISTANCE

We selected a comprehensive set of defense methods that target these three domains to evaluate the defense resistance of 3S-attack. In particular, we consider the following representative defenses: STRIP Gao et al. (2019), Grad-CAM Selvaraju et al. (2017), Fine-Pruning Liu et al. (2018a), and Frequency-based Trigger Detection (FTD) Zeng et al. (2021) where STRIP and Grad-CAM are based on spatial domain; FTD is based on spectral domain; and FP is based on semantic domain. The remainder of this section presents the evaluation of 3S-attack against each of these defenses in detail.

**STRIP** The core idea behind STRIP is that backdoor triggers are typically designed to be highly robust in order to ensure a high attack success rate, whereas benign features in input samples tend to be more fragile and suscepti-ble to disruption. Figure 6 presents the per-formance of STRIP against BadNets Gu et al. (2019) and the proposed 3S-attack on the GT-SRB and Animal10 datasets. Note that Bad-Nets is used solely to demonstrate the effective-ness of the defense against classical backdoor attacks, as well as to show the outcome when

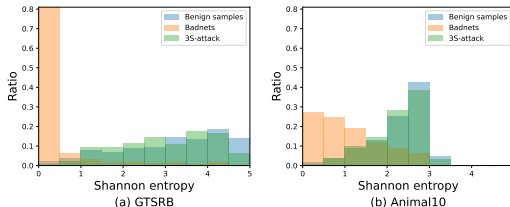

Figure 6: Experimental results of STRIP against Badnets and 3S-attack on GTSRB and Animal10 datasets.

an attack fails to bypass the defense. The results show that STRIP is effective in identifying poi-soned samples in BadNets, as their distribution (orange) significantly diverges from that of benign samples (blue). However, in the case of 3S-attack, the distribution of poisoned samples (green) closely resembles that of benign samples, making them indistinguishable. As a result, no reliable threshold can be set to effectively separate poisoned samples introduced by 3S-attack, allowing it to successfully evade detection by STRIP.

**Grad-CAM** When Grad-CAM is used defen-sively, it produces a heatmap (saliency map) that highlights the regions of an input sample to which the model pays the most attention dur-ing classification. Figure 7 illustrates saliency maps for benign samples (top), poisoned sam-ples from BadNets (middle), and poisoned sam-ples from the 3S-attack (bottom). For benign samples, the model's attention is correctly con-centrated on the main features or objects within the image. However, in poisoned samples gen-erated by BadNets, the model's focus is pre-dominantly on the trigger region, regardless of the true label or semantic content of the image. In contrast, the model's behavior on poisoned

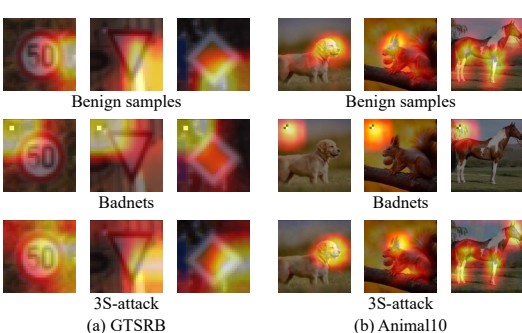

Figure 7: Experimental results of Grad-CAM against Badnets and 3S-attack on GTSRB and An-imal10 datasets.

samples from the 3S-attack closely resembles its behavior on benign samples, with attention dis-tributed over the primary semantic regions. This is because the 3S-attack trigger is embedded using features already associated with the benign class, resulting in no specific spatial region being con-sistently highlighted as the trigger area. As a result, any defenses that is further developed based on Grad-CAM such as Doan et al. (2020) can be bypassed 3S-attack

**Fine-pruning** Fine-pruning (FP) assumes that certain neurons in a backdoored model are primar-ily activated by triggers and remain inactive on benign inputs. By feeding a clean dataset into the model and monitoring neuron activations, consistently inactive neurons are identified as potential backdoor carriers and are pruned or suppressed to disable the attack. Figure 8 presents the results of applying the FP defense to the 3S-attack on the GTSRB and Animal10 datasets. The X-axis is

the ratio of neurons that being deactivated, and the Y-axis is the BA and ASR after such portion of neurons being deactivated.

It is evident that as the pruning rate increases, the benign accuracy (BA) declines more rapidly and earlier than the attack success rate (ASR). This indicates that there is no effective pruning threshold at which ASR is substantially reduced without also significantly degrading the model's performance on benign samples. One explanation is that the 3S-attack embeds the trigger using complex, distributed features that engage a wide range of neurons. As a result, neurons responsible for recognizing benign features and those involved in recognizing the trigger may overlap. This makes it difficult to isolate and remove backdoor-specific neurons without simultaneously impairing the model's normal functionality. Besides, we have observed that BA and ASR have oscillated significantly in experiment on Animal10 dataset. This is caused by the model's limited redundancy on this dataset that when Fine-Pruning removes even a small fraction of these neurons, the model rapidly loses critical feature extractors and suffers an immediate and uncontrolled accuracy drop.

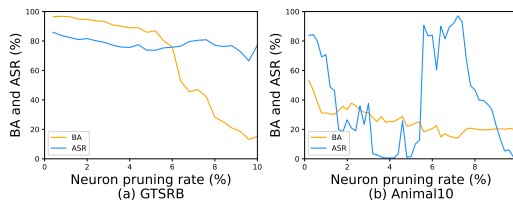

Figure 8: Experimental results of FP defense against 3S-attack on GTSRB and Animal10 datasets.

**Frequency based Defense** Frequency-based Trigger Detection (FTD) uses a dataset constructed with diverse known triggers to train a binary classifier based on spectral features extracted to distinguish benign and poisoned samples. Table 6 shows that FTD performs well in detecting certain types of backdoor attacks and their corresponding triggers. Notably, even when trained on a limited variety of trigger patterns, the FTD detector demonstrates some generalization ability and can successfully detect previously unseen trigger types. However, its effectiveness diminishes when facing attacks like Wanet Nguyen & Tran (2021) and the proposed 3S-attack. This is because the triggers in these attacks exhibit significantly different frequency-domain characteristics compared to those used in the training set. In particular, the 3S-attack modifies only a very small subset of frequency components, making the resulting spectral changes too subtle for the detector to reliably distinguish from benign samples. As a result, the FTD classifier fails to recognize the poisoned samples generated by 3S-attack as anomalous.

|                | Detection Rate (%) | |
| Attack methods | GTSRB | Animal10 |
| --- | --- | --- |
| Benign samples | 98.54 | 100 |
| 3S-attack | **1.46** | **0.98** |
| ISSBA | 100 | 99.16 |
| Wanet | 6.11 | 4.36 |
| Bppattack | 98.87 | 99.44 |
| FIBA | 99.98 | 98.76 |
| Badnets | 100 | 99.08 |

Table 6: Experimental results of FTD defense method against multiple backdoor attack schemes on GTSRB and Animal10 datasets.

## 5 CONCLUSION

In this paper, we proposed 3S-Attack, a novel backdoor attack that achieves stealthiness across spatial, spectral, and semantic domains, to fill the gap that existing attacks have only focused on limited domains. The attack constructs a triple-stealthy trigger by extracting class-relevant features using a preliminary model and Grad-CAM, followed by frequency-domain embedding and pixel-level constraint. 3S-Attack is also the first semantic stealthy attack with no access to the victim model or its training process, making it applicable to more realistic threat scenarios. Extensive experiments demonstrate that our method not only maintains high attack success rates, but also achieves superior imperceptibility across multiple domains, and being harder to detect by existing defenses.

## 6 ETHICS STATEMENT

This paper proposes the 3S-attack and presents a corresponding theoretical analysis, this attack belongs to a class of adversarial techniques targeting AI models. Such attacks may, if misused, lead to potential harm or economic loss by compromising the reliability or confidentiality of machine learning systems. All experiments were conducted on publicly available benchmark datasets containing no personally identifiable information, and no real-world deployment was performed. The purpose of this study is to highlight an important security issue in deep neural networks and to support the development of more robust and trustworthy AI systems. We have adhered to the ICLR Code of Ethics throughout the research and preparation of this work.

## 7 REPRODUCIBILITY STATEMENT

We have taken several steps to facilitate the reproducibility of our results. The section 3 clearly describes the formulation of the 3S-attack, including the motivation and thought process underlying each component. Detailed algorithms and hyperparameter settings are provided in the Appendix. All datasets used in the experiments are publicly available, and their preprocessing pipelines are also documented in the Appendix. Anonymised code of this work is provided in the following link: https://anonymous.4open.science/r/anon-project-3776.

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

# A APPENDIX

## A.1 STATEMENT ON AI TOOLS

Portions of the manuscript, such as grammar refinement, clarity improvement, and minor wording suggestions, were provided by ChatGPT. The model was employed solely as a language-editing tool and did not contribute to the conception of the study, the design of experiments, data analysis, or the generation of scientific conclusions. All intellectual content, methodologies, analyses, and interpretations remain entirely the work of the authors. Every section produced with the assistance of the model was critically reviewed and, where necessary, revised by the authors to ensure accuracy, originality, and compliance with the ethical standards of scholarly publishing. The authors accept full responsibility for the integrity and final content of this article.

## A.2 3S-ATTACK OVERVIEW

To summarize, the proposed 3S-attack follows a three-stage process that enables stealthy and effective backdoor injection by leveraging frequency-domain manipulation guided by semantic features. This design ensures that the poisoned samples remain stealthy across spatial, spectral, and semantic domains while evading multiple defense mechanisms. The entire procedure is illustrated in Algorithm 1, which includes the following components:

---

**Algorithm 1** Trigger Extraction and Poisoned Sample Generation Algorithm of 3S-Attack.

---

**Require:** Clean dataset $\mathcal{D}$, target class $c_t$, frequency selection threshold $\delta$, poison distance ratio $\alpha$, pixel change restriction threshold $\tau$
**Ensure:** Poisoned dataset $\mathcal{D}_{poisoned}$
1: Train model $M$ on $\mathcal{D}$
2: Select sample(s) $x_{trig} \in c_t$
3: $S \leftarrow$ Grad-CAM$(M(x_{trig}))$             ▷ Compute saliency maps
4: $\tilde{x}_{trig} \leftarrow S \odot x_{trig}$
5: $F_{ori} \leftarrow$ DCT$(x_{trig})$
6: $F_{tailored} \leftarrow$ DCT$(\tilde{x}_{trig})$
7: $\mathcal{F} \leftarrow \{f : |F_{ori}(f) - F_{tailored}(f)| < \delta\}$
8: Extract trigger: $\{(f, F_{ori}(f)) \mid f \in \mathcal{F}\}$
9: Sample subset $\mathcal{D}' \subset \mathcal{D}$
10: **for all** $x \in \mathcal{D}'$ **do**
11:      $F_x \leftarrow$ DCT$(x)$
12:      **for all** $f \in \mathcal{F}$ **do**
13:          $F'_x(f) \leftarrow (1 - \alpha) \cdot F_{ori}(f) + \alpha \cdot F_x(f)$
14:      **end for**
15:      $\hat{x} \leftarrow$ IDCT$(F'_x)$
16:      **for all** pixel $p$ in $\hat{x}$ **do**
17:          **if** $|\hat{x}(p) - x(p)| > \tau$ **then**
18:              $\hat{x}(p) \leftarrow x(p) + \text{sign}(\hat{x}(p) - x(p)) \cdot \tau$
19:          **end if**
20:          $\hat{x}(p) \leftarrow$ clip$(\hat{x}(p), 0, 255)$
21:      **end for**
22:      Add $\hat{x}$ to $\mathcal{D}_{poisoned}$
23: **end for**
     **return** $\mathcal{D}_{poisoned} \leftarrow \mathcal{D} \cup \mathcal{D}_{poisoned}$

---

**Trigger Extraction:** A preliminary model is trained to generate Grad-CAM saliency maps for samples in the target class. These maps highlight class-relevant features. By comparing the DCT representations of the original and tailored images, a set of key frequency components is selected as the trigger pattern.

**Poisoned Sample Generation:** A subset of clean data is randomly selected, and their DCT coefficients are selectively modified at the trigger frequencies using linear interpolation between their

original values and those in the extracted trigger. The modified representations are then transformed back into the spatial domain via inverse DCT.

**Pixel Value Restriction:**   To preserve visual imperceptibility, the pixel-level differences between each poisoned sample and its original are clipped to a specified threshold.

### A.3   THEORETICAL AND EXPERIMENTAL PROOF OF SEMANTIC SIMILARITY

A core component of 3S-Attack is the use of a preliminary surrogate model to approximate the semantic behavior of the victim model. Although the attacker cannot access the victim's training process or parameters, our method relies on the observation that supported by both theoretical reasoning and empirical evidence: Two independently trained models on similar data tend to focus on similar semantic regions when classifying the same sample.

**Theoretical Justification**   Let $f_{pre}$ and $f_{vic}$ denote the preliminary and victim models, respectively, and let $A_{pre}(x)$ and $A_{vic}(x)$ be their Grad-CAM saliency maps for an input $x$ belongs to class $c$. Grad-CAM computes the spatial importance of each location via:

$$A(x) = \text{ReLU}\left(\sum_k \alpha_k F^k(x)\right), \qquad \alpha_k = \frac{1}{Z}\sum_{i,j}\frac{\partial y_c}{\partial F_{ij}^k(x)}.$$

where $F^k(x)$ is the $k$-th feature map at the last convolutional layer. A key property of CNN classifiers trained to high accuracy is that they must rely on object-relevant features rather than background artifacts. Formally, if both models satisfy:

$$\Pr[f_{\text{pre}}(x) = c \mid x \in c] \approx 1, \qquad \Pr[f_{\text{vic}}(x) = c \mid x \in c] \approx 1.$$

And to do this, their optimal discriminative features must approximate the true object-support region $\Omega_c \subseteq \{1,\dots,H\} \times \{1,\dots,W\}$. Because for a large amount of samples belonging to the class $c$, their only common point is that every image contains the object described by class $c$, which makes extracting the semantic feature that all the images in class $c$ being the only way to achieve a high benign accuracy and generalizability. Thus, although architectures and training data may differ, both models satisfy:

$$\text{supp}(A_{\text{pre}}(x)) \approx \Omega_c \approx \text{supp}(A_{\text{vic}}(x)).$$

which provides the theoretical basis for using $A_{pre}(x)$ as a surrogate for the victim model's semantic focus.

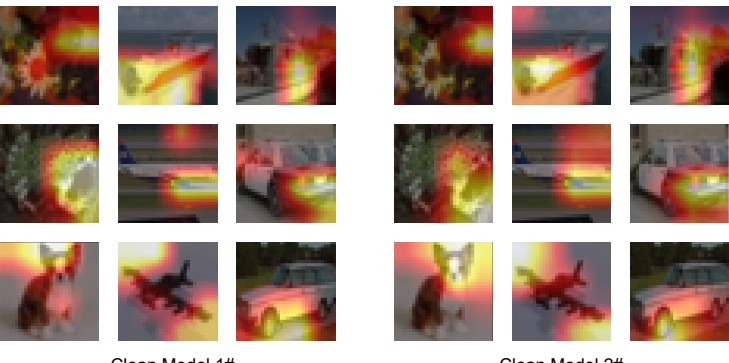

Clean Model 1#                         Clean Model 2#

Figure 9: Saliency map of two clean model classifying same group of samples trained on separated datasets.

**Experimental Validation**   We designed a simple yet effective experiment to prove the above logic. We take a clean dataset, for example CIFAR-10, and randomly split this dataset into two disjoint subsets where each sample in the original CIFAR-10 can and only can belongs to one of the subsets, which we call $subset_1$ and $subset_2$. Then both subset are used to train a model independently to

acquire $model_1$ and $model_2$. Then the samples in test set that never existed in $subset_1$ and $subset_2$ are computed saliency map via Grad-CAM, when we compare the saliency map generated by both models on the same samples. The results are shown in Figure 9, from which we can tell that the semantic important region that two models identified on same sample are similar. This indicate that despite trained on different datasets, different models that all achieved acceptable classification accuracy on the same class, can all focus on the roughly correct region when processing same samples in this class. Because the test samples are never seen during the training of either model, any similarity in their saliency maps must arise from shared semantic structure rather than memorization. Consequently, a preliminary model trained on attacker-owned dataset can reliably approximate the victim model's semantic focus for the target class.

### A.4 WHY STABLE DCT COMPONENTS REPRESENTS SEMANTIC FEATURE

**Why Selecting Stable DCT Components**   A central motivation for selecting DCT components whose magnitudes change minimally between the original image and its Grad-CAM weighted counterpart lies in the way semantic information is preserved under spatial masking. Grad-CAM attenuates non-discriminative regions while retaining the object-related structure that the model relies on for classification. Because a trained deep network functions as a semantic feature extractor, its Grad-CAM map reveals the regions that encode the class-defining content, even when these semantics are difficult to characterize explicitly.

Given that the DCT is a linear transform, suppressing background pixels produces predictable changes only in the frequency components associated with the removed background content. In contrast, the components encoding the preserved semantic structure remain largely stable. Consequently, the DCT components that exhibit small magnitude differences before and after Grad-CAM weighting correspond to model-dependent semantic features of the target class. By selecting these stable components, 3S-Attack isolates spectral patterns that reflect object-level semantics while filtering out background or spurious cues.

Our ablation study further validates this interpretation: Replacing the stability-based selection with random frequency selection leads to a marked reduction in semantic stealth and significantly increases activation-space separability. This provides empirical evidence that stability under Grad-CAM masking effectively identifies the spectral components responsible for semantic preservation. Therefore, we have the following reasoning chain that leads to the methodology of 3S-attack: Grad-CAM $\rightarrow$ preserve semantic region $\rightarrow$ spatial masking $\rightarrow$ linear DCT $\rightarrow$ stable frequencies $\rightarrow$ semantic meaningful frequencies.

**Determining Frequency Selection Threshold Value**   Besides, we cannot determine, for any given sample, how many, which, or what kinds of features represent the content described by its label, nor to what extent these features reflect that label. Applied to the trigger extraction pipeline proposed in this paper, this means that there is currently no explicit theoretical derivation that can guide us in identifying the optimal values or appropriate ranges of the parameters in 3S-attack. Consequently, the selection of parameter values is largely empirical.

For the frequency selection threshold in particular, our reasoning is as follows: We know for each frequency, the degree of difference between the original image and the tailored image, and that the tailored image preserves the parts the model considers most important for classifying the sample. Therefore, frequencies that exhibit smaller differences correspond to more important features of the sample. In this way, we obtain the importance of each frequency in reflecting the semantic content associated with the sample's label. However, we still lack another key piece of information, namely how many of the top-ranking frequencies are sufficient to capture the sample's semantic features, i.e. where the frequency selection threshold should be set. We thus adopt an empirical approach, tuning this parameter to balance the effectiveness, stealthiness, and defense resilience of the 3S-attack. The experimental results in Figure 5 show that 3S-attack can achieve strong performance across a relatively wide parameter range. Therefore, even when attackers have no knowledge of the victim-side configuration, they may choose parameter settings that cause smaller perturbations to the sample, while still retaining a high probability of successfully implanting a backdoor into the victim model.

A.5 TIME EXPENDITURE FOR PREPARATIONS

In 3S-attack, before the attack can perform the attack, prepositive steps are required, including training a preliminary model, apply Grad-CAM to compute saliency map and extract spectral trigger, and generate poison samples. In this section, we evaluate and report the time expenditure of each step.

**Train Preliminary Model**   As described in the paper, the preliminary model only needs to produce Grad-CAM saliency maps that roughly localize the dominant object region, it is not required to match the architecture, depth, or accuracy of the victim model. In our experiments, a CNN smaller than ResNet/VGG/WRN is sufficient as preliminary model. Training such a model takes only a small fraction of the time needed to train a standard classifier on the same dataset, not only because the parameter amount is smaller than mainstream models, but also because epochs required during training is much smaller. In experiments, typically 10-50% of the victim model's training time is enough to train a well functioning preliminary model, depends on the exact parameter amount of the preliminary model and victim model.

**Generate Saliency Map and Poison Samples**   Once the preliminary model is trained, the subsequent operations including Grad-CAM computation, DCT transform, frequency selection, and trigger injection are purely feed-forward computations. The poison sample generation process completes in seconds to a few minutes, depending on the sample resolution, number of poison samples, and hardware details. Thus, the dominant cost is not data poisoning, but the one-time preliminary model training.

**Time Constrain for Attacker**   In the data-poisoning threat model we study, the attacker can prepare poisoned data long before the victim trains any model, since there is no time constraint or interaction requirement. Therefore, even if the attacker chose to train several preliminary models or refine the trigger multiple rounds, this cost remains entirely offline and does not affect the success or practicality of the attack. Besides, once a trigger is extracted for a given target class, it can be reused for different victim models, different training runs, and even different datasets of the same class semantics. This significantly reduces the amortized cost of the attack in practice.

A.6 STEALTHINESS IN SEMANTIC DOMAIN

To evaluate the semantic stealthiness of 3S-attack, we examine how closely the neuron activation patterns elicited by poisoned samples resemble those of benign samples from the target class. We conduct this analysis on the CIFAR-10 dataset using a ResNet-18 model. Specifically, benign samples from the target class and poisoned samples generated by different attacks are grouped into two subsets. Each sample is passed through the backdoored model, and activation vectors from the second last layers are collected. These activations naturally form empirical distributions for the benign subset and the poisoned subset, respectively. To quantify their similarity, we compute the squared Maximum Mean Discrepancy ($MMD^2$), a widely adopted metric for comparing distributions of high-dimensional neuron activations. The value of $MMD^2$ ranges from 0 to 2, with smaller values indicating greater similarity.

Table 7 summarizes the results. To contextualize the scale of this metric, we additionally evaluate two baseline scenarios. First, we randomly divide benign samples from the same class into two subsets and compute their $MMD^2$; as expected, the resulting score (first row in in table) is near zero, confirming that semantically consistent samples yield nearly identical activation distributions. Second, we compute $MMD^2$ between benign samples drawn from two different classes, which produces values close to the upper bound (second row in in table), reflecting that the underlying activation patterns are largely independent. For

| Attack methods | $MMD^2$ score |
|---|---|
| Same class | 0.0004 |
| Diff classes | 1.9801 |
| 3S-attack | **0.5996** |
| ISSBA | 1.2372 |
| Wanet | 1.0137 |
| Bppattack | 1.0283 |
| FIBA | 0.8946 |
| Badnets | 1.4828 |

Table 7: $MMD^2$ score of 3S-attack compared with other baseline attacks and specific situations

most baseline backdoor attacks, the activation distributions of poisoned samples diverge significantly from those of benign target-class samples, with $MMD^2$ typically exceeding 1. This indicates that unless intentionally enhance the stealthiness in semantic domain , backdoor attacks can leave discernible semantic footprints inside the model. In contrast, 3S-attack attains an $MMD^2$ of approximately 0.6—substantially lower than competing methods—demonstrating that poisoned samples produced by our method activate neurons in a manner much closer to the benign target-class distribution. While the performance is not as ideal as methods that explicitly enforce semantic alignment by modifying the training process or introducing additional losses, 3S-attack achieves significantly stronger semantic stealthiness under a more realistic threat model where the adversary cannot influence model training.

### A.7    3S-attack Under Consistent Model Structure

The proposed 3S-attack is designed with the idea of being effective across datasets and model structures instead of relying on any specification. The experiment results in Section 4.2 demonstrated the effectiveness of 3S-attack deployed to a variety of datasets and models. To further evaluate the consistency of effectiveness, we conducted the following experiments. Instead of selecting different model structure for each dataset, we deploy a unified ResNet-18 to dataset GTSRB, CIFAR-10, CIFAR-100, Animal-10, and Imagenet to evaluate the per-

|          | Clean | 3S-attack | | | |
|----------|-------|-------|-------|-------|-------|
| Dataset  | BA    | BA    | ASR   | PSNR  | SSIM  |
| GTSRB    | 98.36 | 96.55 | 94.12 | 32.78 | 0.979 |
| CIFAR-10 | 86.40 | 84.65 | 89.29 | 35.65 | 0.969 |
| CIFAR-100| 54.35 | 52.13 | 92.78 | 31.14 | 0.943 |
| Animal-10| 88.08 | 87.32 | 97.42 | 30.83 | 0.962 |
| Imagenet | 74.80 | 73.10 | 86.26 | 32.32 | 0.963 |

Table 8: The performance of 3S-attack under unified ResNet-18 model structure on various datasets.

formance of 3S-attack. Note that due to the input and output structure differences of each dataset, minor adjustments to the ResNet-18 are inevitable, we can only ensure the main structure is consist across experiments.

The results are shown in Table 8, from which we can tell that 3S-attack maintains stable performance across all datasets when the underlying model structure is fixed. Compared with Table 2, the benign accuracy after poisoning remains within a small margin of the clean model's accuracy, demonstrating that using a unified architecture does not compromise the attack's stealthiness or its impact on the primary classification task. At the same time, the attack success rate consistently achieved the same level compared with other attacks across all datasets, indicating that the principle of 3S-attack generalizes well under architectural homogeneity. Moreover, the PSNR and SSIM values remain nearly identical as that in Table 3 for all datasets, showing that the attack performance is universal across datasets. This consistent performance confirms that the proposed 3S-attack does not rely on model-specific properties. Rather, its effectiveness stems from exploiting stable semantic and spectral patterns that persist across architectures.

### A.8    Possible Defense

Apart from the above defenses that 3S-attack can bypass, we also explored what defenses might be effective on detecting and defending against the 3S-attack. The following two defense methods are found to some extent, effective against the 3S-attack.

**Neural Cleanse**    The core idea behind Neural Cleanse (NC) Wang et al. (2019) is based on the observation that attackers typically aim to design triggers as small and inconspicuous as possible. Moreover, backdoor-ed models often rely on a few key pixels from the trigger pattern to cause misclassifications. As a result, for the target class, it is usually possible to identify a small trigger pattern that, when attached to a wide range of benign inputs, consistently causes misclassification into that class. In contrast, for clean (non-target) classes, any synthesized *trigger* that causes benign samples to be misclassified into those classes tends to be much larger, as there is no actual backdoor associated with them. By reverse-engineering potential *triggers* for all classes and comparing their sizes, NC identifies the class with an abnormally small trigger as the likely backdoor target.

Figure 10 presents the anomaly index for each class in the NC defense applied to a 3S-attack where the target class is *7*. Subfigure (a) shows the results on the GTSRB dataset, where class *7* exhibits

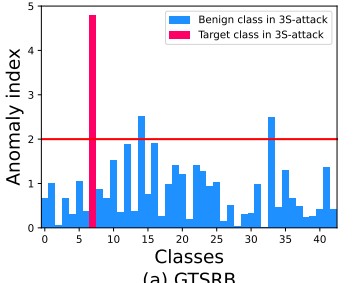 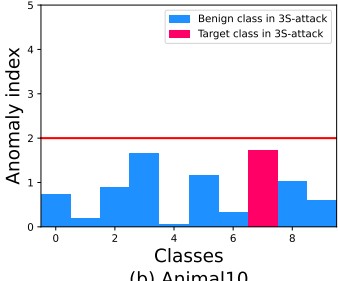

Figure 10: Experimental results of Neural Cleanse against 3S-attack on GTSRB and Animal10 datasets.

a significantly higher anomaly index, indicating that the 3S-attack is effectively detected, although some other class are also flagged as false positive. However, in subfigure (b), based on the Animal10 dataset, the anomaly index of class 7 is 1.73—still relatively high but below the threshold. Moreover, another clean class also has a comparable anomaly index of 1.65. These results suggest that while NC is effective in detecting the 3S-attack under certain conditions, its reliability is not guaranteed across all settings. Due to the black-box nature of DNNs, the underlying reasons for this inconsistency are difficult to pinpoint. One possible explanation is that, in some cases, the perturb introduced by poison sample generation process is not sample-specific enough that resulted in trigger pattern in each specific sample still have some common pattern in spatial domain. Therefore the model learns to associate this certain subtle, recurring pixel patterns as the effective trigger, thereby enabling successful reverse engineering by NC.

**Activation Clustering**    The idea behind Activation Clustering (AC) Chen et al. (2018) is similar to that of Fine-Pruning, in that certain neurons in a backdoored model—particularly those in the fully connected layers—are specifically responsible for recognizing the presence of a trigger. As a result, although poisoned and benign samples from the target class may yield the same prediction, the internal mechanisms differ, as they activate different subsets of neurons. Based on this observation, for each class in the model, one can collect neuron activation patterns and apply clustering analysis. If the activations naturally separate into two distinct clusters, it is likely that the class is a backdoor target; otherwise, the class is considered benign.

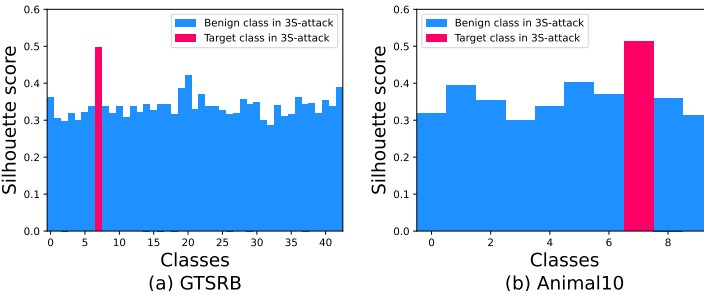

Figure 11: Experimental results of Activation Clustering against 3S-attack on GTSRB and Animal10 datasets.

Figure 11 illustrates that AC is effective against the 3S-attack across different datasets, as the Silhouette scores of the target class are consistently higher than those of benign classes. This may be attributed to the fact that, although 3S-attack is designed to make poisoned samples activate similar neurons as benign ones, the internal optimization process of the target model remains a black box and is beyond the attacker's control. Consequently, some neurons may still be implicitly assigned the task of recognizing trigger-specific patterns. These findings suggest that AC is a particularly strong defense that designing a backdoor attack capable of evading AC without access to the model training process remains an extremely challenging task.

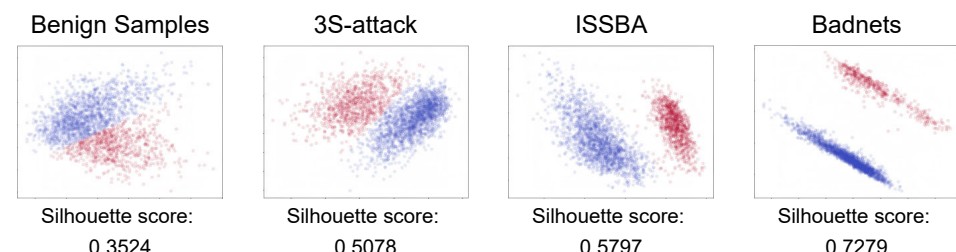

Figure 12: The activation distribution map of clean class, target class in 3S-attacks, and target class in other attacks and the corresponding Silhouette score.

However, in fact, while the target class in 3S-attack achieved a Silhouette score around 0.5, it is still much better than other attacks since they usually result in a Silhouette score around 0.6 to 0.9. Figure 12 visualized the distributions of samples in clean and poison classes under various attacks, here we take ISSBA Li et al. (2021b) and Badnets Gu et al. (2019) for example. Existing backdoor attacks that not built to be semantically stealthy is quite visible on the ICA/PCA precessed maps. In ISSBA and Badnets, it is clear that the dots are suitable for two clusters, the associated Silhouette score also indicating they can be easily detected by AC defense. However, the activation of 3S-attack and benign samples looks more like the appearance of that of benign classes, the corresponding Silhouette score is also lower than existing attacks. Therefore, although this study has not yet succeeded in completely bypassing the AC defense without access to the model training process, it nevertheless shows promising prospects for achieving this goal in the future.

## A.9 DISCUSSION

In this section, we analyze the key findings from the experiments, compare 3S-attack with existing works, identify limitations, and discuss potential future directions.

**Contributions and Impact** This work is the first to propose a backdoor attack that is simultaneously stealthy in spatial, spectral, and semantic domains. Furthermore, it achieves semantic stealthiness without requiring access to the model training process—an important advancement for practical black-box attacks. These findings imply that backdoor attacks can remain effective even under strong stealth constraints, underscoring the considerable potential for advancement in the design of both backdoor attacks and corresponding defenses.

**Core Properties** The experimental results demonstrate that 3S-attack is a feasible, stealthy, robust, and defense-resistant backdoor attack. It achieves consistently high ASR across datasets of varying complexity and resolution, including MNIST, GTSRB, CIFAR-10/100, and Animal-10, confirming its general feasibility. Meanwhile, the attack induces only minimal perceptual distortion, as evidenced by high PSNR and SSIM scores—often exceeding all baseline methods. This validates its spatial and perceptual stealthiness.

**Hyperparameter and Model Robustness** The 3S-attack remains stable across a wide range of parameters, including poison rate, frequency threshold, poison distance ratio, and pixel-level restriction. Even under conservative configurations, 3S-attack retains high effectiveness, showing robustness to hyperparameter variations. Moreover, it generalizes well across different model architectures, from simple CNNs to deep residual networks, further enhancing its applicability.

**Defense Resistance** Several defense mechanisms are rendered ineffective against 3S-attack. STRIP fails to detect poisoned samples due to overlapping entropy distributions between benign and poisoned samples are close. Grad-CAM-based detection is also evaded because Grad-CAM consistently highlights natural areas, even in poisoned samples. As a result, not only Grad-CAM but also its derivative defenses—such as saliency-based trigger localization—are effectively bypassed.

**Failure of FTD** FTD is designed to detect spectral anomalies but fails against 3S-attack. As shown in Figure 2, the trigger typically occupies only 1%–5% of the frequency map and lacks any

structured or localized pattern. This seemingly randomness prevents the FTD classifier, trained on known triggers with regular frequency characteristics, from generalizing to 3S-attack. Consequently, FTD consistently misclassifies 3S-poisoned samples as benign.

**Partial Detection by NC and AC**   Despite its stealth, 3S-attack remains partially detectable by Neural Cleanse (NC) and Activation Clustering (AC). NC succeeds in dataset GTSRB, where class patterns are constrained, but fails on Animal-10 due to semantic complexity.  While AC is more robust that although 3S-attack aligns poisoned inputs with benign attention maps, it cannot fully eliminate discrepancies in deep-layer activations. These latent differences remain cluster-able, suggesting that 3S-attack does not yet achieve complete semantic stealthiness.

**Limitations and Future Work**   An area where the 3S-attack could be further improved is its stealth at the feature (semantic) level.  Specifically, Activation Clustering can still detect subtle activation differences between benign and poisoned samples. Enhancing semantic invisibility without access to model internals remains a difficult but essential direction. Future work may explore: (1) adaptive frequency selection strategies, (2) activation-aligned poisoning to evade AC, and (3) extending the attack to more complex modalities such as video, text, and multimodal learning.

**Summary**   3S-attack demonstrates that multi-domain stealth is both achievable and effective.  It exposes critical vulnerabilities in current AI systems and motivates the design of more advanced defense strategies.

