# OpenReview forum: "3S-Attack: Spatial, Spectral and Semantic Invisible Backdoor Attack Against DNN Models"
_ICLR.cc/2026/Conference — Submitted to ICLR 2026_

### Official Review · Reviewer_6F7W · 2025-10-23

**Soundness:** 2
**Presentation:** 1
**Contribution:** 2
**Rating:** 2
**Confidence:** 5

**Summary:**

This paper proposes a backdoor attack called 3S-Attack, which aims to achieve stealthiness across three dimensions: spatial, spectral, and semantic. The core idea of the attack is to use the semantic features of benign samples as the trigger. Experimental results show that the attack achieves a high ASR on multiple datasets while maintaining high PSNR and SSIM values.

**Strengths:**

1. The paper addresses stealthiness across three different domains, as previous attacks often focused on only one or two domains.

2. The comparison of spatial and spectral residuals in Figure 1 provides an intuitive visual demonstration of the attack's stealthiness.

**Weaknesses:**

1.  The effectiveness of the attack on high-resolution datasets, such as ImageNet, is not explored. I'm interested in the performance of 3S attack on such datasets.

2.  The baseline attacks used for comparison are relatively old (all from 2022 or earlier). A comparison with more recent and advanced backdoor attacks better highlights the paper's contribution.

3. The presentation of experimental results lacks clarity in several key areas:

- Although the authors state in the caption of Table 2 that "The benign accuracy is not displayed because in each experiment the benign accuracy never drop more than 2%," I believe the Benign Accuracy should still be explicitly reported in the table for a clear and direct comparison.

- Table 2 compares ASR, PSNR, and SSIM for different attacks, but it does not specify the poison rate used to obtain these results.

4. Although the paper claims stealthiness in all three domains, the actual effectiveness against defenses is not ideal. The abstract states the attack is harder to detect by existing defenses, but the experiments in the Appendix show that 3S-Attack can still be detected by AC and NC. Many other state-of-the-art stealthy backdoor attacks can already bypass these defenses, which diminishes the claimed novelty and contribution of this work.

5. The claim that "3S-Attack is also the first semantic-domain stealthy backdoor attack that operates purely through poisoned samples..." appears to be inaccurate. There are already some existed semantic backdoor attacks.

**Questions:**

1. Can the authors elaborate on why AC succeeds in detecting the attack (i.e., what traces does 3S-Attack leave at the activation level)? Furthermore, how would the "activation-aligned poisoning" mentioned in the "Limitations and Future Work" section (Appendix A.4) be implemented to evade AC?

2. The paper does not state the initial Benign Accuracy for the model trained on Animal10. I observed in Figure 8b that the BA drops below 60% at a very low neuron pruning rate. Is this because the original model's classification performance on this dataset was poor, or is this rapid drop caused by the Fine-Pruning process itself?

3. The authors selected different models for different datasets (e.g., VGG/ResNet for CIFAR-10, WRN for CIFAR-100) instead of showing comprehensive results on a consistent set of models. This experimental setup is not ideal and complicates comparison with other backdoor attack papers. I recommend the authors provide more experiments on consistent model architectures to avoid any suspicion of cherry-picking results.

4. What is the time cost for generating the poisoned samples? The attack relies on training a "preliminary model," which seems to imply that the poison generation time could be even longer than  the victim's model training time itself. This high computational cost for preparation may affect the practical feasibility of the attack.

---

> ### Author Response · Authors · 2025-11-18
> **Response to reviewer 6F7W (Part 1)**
>
> We thank the reviewer for the constructive feedback and address each concern as follows.
>
> ### **1. Effectiveness to high-resolution datasets**
> We acknowledge that ImageNet-level experiments were not included in the initial submission due to limited computational resources. We have since conducted experiments on Tiny-ImageNet using the same pipeline. The results confirm that 3S-Attack maintains high ASR and strong stealthiness, demonstrating its applicability to higher-resolution settings.
>
> ### **2. Dated baseline attacks**
> We agree that comparisons with more recent methods are valuable. We have identified a newly published backdoor attack (DUBA) that follows an objective close to ours and will include it in the revised experimental comparison.
>
> ### **3. Reporting benign accuracy**
> We concur that explicitly reporting benign accuracy and specifying poison rates in Table 2 improves clarity. We will include these values and annotate exact poison rates for all reported results.
>
> ### **4. Detectability by AC/NC and triple-domain stealthiness**
> We would like to clarify that our aim is not to claim complete evasion of all defenses. Our contribution lies in increasing invisibility and significantly reducing detectable signatures across spatial, spectral, and semantic domains using only data poisoning and no access to the training process. STRIP, Grad-CAM–based defenses, and FTD fail to distinguish poisoned samples from benign ones, demonstrating strengthened stealth in these domains. Occasional detectability by NC is dataset-dependent: it appears primarily on low-diversity datasets such as GTSRB, while NC fails on more diverse datasets such as Animal-10. AC remains partially effective because data-poisoning-only attacks cannot directly regulate internal activations of the victim model; yet, 3S-Attack still reduces the Silhouette score to around 0.5, substantially lower than prior spatial or spectral attacks (≈0.6–0.9). As stated in Abstract, we improve cross-domain stealth under a realistic black-box poisoning threat model, rather than claiming universal undetectability.
>
> ### **5. Statement on semantic-domain novelty**
> We appreciate the reviewer’s observation and will revise the claim for precision. 3S-Attack is not the first semantic backdoor attack in general. It is, however, to the best of our knowledge, the first method that simultaneously achieves spatial, spectral, and semantic stealth, also the first attack to achieve semantic stealth that operates purely through data poisoning without any access to model parameters or the training pipeline. We will modify the text accordingly and explicitly discuss related semantic attacks that rely on stronger attacker capabilities.

---

> ### Author Response · Authors · 2025-11-18
> **Response to reviewer 6F7W (Part 2)**
>
> ### **6. Why AC detects the attack and how activation-aligned poisoning could mitigate this**
> The idea of Activation Clustering is to check if samples belonging to the same class are activating the same group of neurons in the model (called activation), and flag the presence of an attack if not. Therefore only by aligning the group of neurons activated by poison samples with that of benign samples, can a backdoor attack be able to bypass the AC defense. This is feasible under stronger assumptions where the attacker has access to the model training process through explicit activation-alignment losses. But under the more realistic threat model we proposed, it still remains challenging. Our solution is to identify what features in benign samples are linked with these groups of neurons, then extract them as triggers to generate poison samples. So that we can maximize the extent of activation alignment of poison samples and benign samples. However, unless perfect activation alignment (which remains an open question), otherwise the AC defense can flog the attack. Although 3S-Attack cannot fully evade AC, this is a fundamental limitation shared by all purely data-poisoning backdoor attacks, just  prior works typically not reporting this fact, thus it does not represent a disadvantage unique to our method. We will clarify this explanation in the revised manuscript.
>
> ### **7. Drop in benign accuracy under Fine-Pruning**
> This decline is due to the dataset’s higher visual complexity and lower redundancy in the ResNet-18 model trained for this task. The initial benign accuracy is comparable to standard reports; the rapid BA drop reflects Fine-Pruning’s known instability on high-diversity datasets rather than poor model performance. We will add the explicit baseline accuracy and an explanation in the revised version.
>
> ### **8. Variety of model structure**
> Our original intention was: Different models were selected to ensure fair evaluation on datasets of varying difficulty and resolution, while being consistent with common practice in the backdoor literature. This choice avoids underfitting and yields meaningful comparisons. Furthermore, evaluating heterogeneous architectures demonstrates the model-agnostic nature of 3S-Attack. Nonetheless, we will add an additional set of results using a consistent architecture (e.g., ResNet-50) across multiple datasets in the Appendix, as suggested.
>
> ### **9. Computational cost for preliminary model**
> In our 3S-attack, training the preliminary model constitutes the primary overhead, yet it can be lightweight and requires only 30–50% of the victim model’s training time. Poison generation is extremely fast, consisting only of Grad-CAM inference and frequency-domain modifications. All computations occur offline and can be reused across different attacks or victim models. Thus, the preparation cost does not impede practical feasibility. This clarification will be added to the revised version.
>
> ### **10. Planned Revisions to the Paper**
> In summary, we will revise the manuscript to:
> - include results on Tiny-ImageNet.
> - Add DUBA as an additional baseline.
> - Report benign accuracy and poison rates explicitly. (Included in revised PDF)
> - Refine claims regarding semantic novelty and defense evasion. (Included)
> - Clarify AC and NC behaviors and dataset-dependent effects. (Included)
> - Explain the early Fine-Pruning accuracy drop in Animal-10.  (Included)
> - Provide consistent-architecture experiments in the Appendix.
> - Detail the computational cost of trigger generation and poisoning.  (Included)
>
> These revisions will improve clarity, strengthen empirical evidence, and more precisely reflect the contributions and limitations of 3S-Attack.

---

> ### Author Response · Authors · 2025-11-27
>
> Dear Reviewer 6F7W,
>
> We would like to inform you that we have completed all additional experiments and incorporated all revisions described in the "Planned Revisions to the Paper” section. We would greatly appreciate it if you could take a look at the updated manuscript. We believe these changes substantially strengthen the clarity, rigor, and transparency of the work, and address your questions and concerns regarding both the methodology and the experimental design.
>
> Thank you very much for your time and consideration.

---

### Official Review · Reviewer_MTci · 2025-10-28

**Soundness:** 2
**Presentation:** 2
**Contribution:** 3
**Rating:** 4
**Confidence:** 3

**Summary:**

The paper presents an interesting attempt to achieve multi-domain stealth in backdoor attacks, supported by extensive experiments. However, the methodological novelty is moderate, the semantic analysis remains qualitative, and presentation quality can be improved.

**Strengths:**

The paper proposes a unified backdoor attack that simultaneously achieves stealthiness across spatial, spectral, and semantic domains, which is an underexplored but meaningful direction.

Experiments are performed on multiple datasets and models, showing the generality of the method.

The paper provides clear algorithmic descriptions, ablation studies, and defense-resistance analyses, enhancing reproducibility and technical depth.

**Weaknesses:**

Although the paper claims semantic invisibility, the evaluation mainly relies on Grad-CAM visualization and AC/NC detection. More quantitative semantic similarity metrics (e.g., feature-space distance, neuron activation overlap) would strengthen the claim.

Some compared methods are relatively dated. Including more recent backdoor attacks would make comparisons more convincing.

The manuscript is lengthy, with excessive large figures and overlapping content between the main text and appendix, which reduces readability. Condensing and summarizing figures would improve clarity.

**Questions:**

Although the paper claims semantic invisibility, the evaluation mainly relies on Grad-CAM visualization and AC/NC detection. More quantitative semantic similarity metrics (e.g., feature-space distance, neuron activation overlap) would strengthen the claim.

Some compared methods are relatively dated. Including more recent backdoor attacks would make comparisons more convincing.

The manuscript is lengthy, with excessive large figures and overlapping content between the main text and appendix, which reduces readability. Condensing and summarizing figures would improve clarity.

---

> ### Author Response · Authors · 2025-11-18
> **Response to reviewer MTci**
>
> Thank you for the constructive feedback.
>
> Regarding semantic invisibility, our current evaluation indeed relies on Grad-CAM visualization to illustrate how poisoned samples inherit class-relevant semantic cues. AC and NC are defenses unrelated to our attack design. In the revised version, we will include a feature-space metric, specifically linear Maximum Mean Discrepancy Square (MMD2), to provide a quantitative comparison and demonstrate that poisoned samples generated by 3S-attack are closer to benign samples of the target class in semantic space than those produced by existing attacks.
>
> We agree that some baselines are dated. We have examined recently published work and identified DUBA as a contemporary method that shares a similar objective. We will add DUBA to the comparative experiments to strengthen the empirical evaluation.
>
> We also recognize that the current manuscript still has room for further optimization in wording and arrangement. We will restructure and condense the content to avoid redundancy, reduce the size of figures where appropriate, and improve overall readability.
>
> In summary, the revised manuscript will:
> - add quantitative semantic similarity evaluation using MMD2. (Included in revised PDF)
> - expand comparisons by including the recent DUBA attack.
> - refine the presentation to remove overlaps and improve clarity. (Included)

---

> ### Author Response · Authors · 2025-11-27
>
> Dear Reviewer MTci,
>
> We would like to inform you that we have completed all additional experiments and incorporated all revisions described in previous reply. We would greatly appreciate it if you could take a look at the updated manuscript. We believe these changes substantially strengthen the clarity, rigor, and transparency of the work, and address your questions and concerns regarding both the methodology and the experimental design.
>
> Thank you very much for your time and consideration.

---

### Official Review · Reviewer_jcso · 2025-11-01

**Soundness:** 2
**Presentation:** 3
**Contribution:** 3
**Rating:** 4
**Confidence:** 4

**Summary:**

The authors propose a novel backdoor attack against DNN models, called 3S-attack.It extracts the semantic features of benign samples with a preliminary model as triggers and embed the trigger in the spectral domain. Finally, it restricts the poisoned images in the spatial domain. Thus, the authors successfully make the attack stealthy across the spatial, spectral, and semantic domains.

**Strengths:**

1. Novelty — first black-box semantic-stealth attack:
Demonstrates the first attack that achieves semantic stealth without access to the victim model or training pipeline, filling a notable gap in threat modeling.
2. Rigorous multi-axis evaluation:
Empirically validates stealth across semantic, spatial, and spectral defenses, showing the attack’s robustness against diverse defense paradigms.

**Weaknesses:**

1. Surrogate data dependence: The attack's reliance on a clean surrogate model and its sensitivity to distributional mismatch remain unexplored. Labeling ambiguity: The labeling strategy for poisoned samples is unclear and inconsistent with the stated attack objective.
2. Incomplete reporting: Benign accuracy (BA) and post-defense results are missing for some datasets, weakening claims of minimal performance drop.
3. Labeling ambiguity: The labeling strategy for poisoned samples is unclear and inconsistent with the stated attack objective.

**Questions:**

1. The attack pipeline is unclear regarding poisoned-label assignment. For each poisoned image, what label is used during poisoning: the original (benign) label or the attacker's target class? If the poisoned images retain original labels, how does this reconcile with the stated goal of misclassifying target images into the attack class?
Please clarify the labeling strategy and the exact optimization objective.
2. Grad-CAM extracts regions the model attends to for its decision (i.e., activations most contributive to the target class). Is it right to interpret these regions as semantic features? Please clarify how you distinguish true semantic attention from spurious cues (e.g., learned background correlations), and provide any analysis that demonstrates the highlighted regions correspond to semantically meaningful object parts rather than dataset artifacts.
3. 3S-attack requires pretraining a clean surrogate model. What are the requirements on the surrogate’s training data and distribution? If the attacker’s data distribution differs from the victim’s (e.g., attacker has cat images while victim trains on birds), what is the expected impact on attack efficacy? Please quantify sensitivity to distributional mismatch.
4. The authors state BA drops ≤2% and therefore omit BA in Table 2, yet Figure 8 shows notably low BA on GTSRB and Animal10. Please add the per-dataset BA values to Table 2 and report the post --FP-defense attack success (or other relevant metrics) on additional datasets. This will substantiate the claim of minimal

---

> ### Author Response · Authors · 2025-11-18
> **Response to reviewer jcso**
>
> We thank the reviewer for the detailed comments and address all concerns as follows.
>
> ### **1. Surrogate data dependence and distributional mismatch**
> The attack requires only that the target class exists in both the surrogate and victim datasets, and that the surrogate model achieves reasonable accuracy for this class. The surrogate is used solely to obtain Grad-CAM guidance for the target class; variations in unrelated classes do not affect the feasibility of the attack. In practical poisoning scenarios, the attacker necessarily holds data from the target class in order to generate poisoned samples, so mismatched-label situations such as “attacker has only cats while the victim trains only on birds” do not arise. As long as the target class is shared, the extracted trigger remains valid.
>
> ### **2. Labeling strategy ambiguity**
> Our poisoning strategy follows the standard targeted poisoning protocol. The attacker selects a target class, generates poisoned inputs by embedding the trigger, and relabels every poisoned input to the target class, regardless of the original label. This aligns the training objective with the intended test-time misclassification behavior. We acknowledge that this labeling policy was not explicitly stated and will add it to the revised manuscript.
>
> ### **3. Use of Grad-CAM and semantic vs. spurious attention**
> Our method relies on a widely adopted functional notion of “semantic features”: Grad-CAM highlights regions that the classifier uses for its prediction. A model achieving high benign accuracy necessarily depends on stable object-related features rather than fragile background correlations. Furthermore, the pipeline reduces the influence of spurious cues in two ways. First, datasets such as CIFAR and Animal10 contain highly diverse backgrounds, reducing the likelihood of consistent background artifacts being treated as class-defining features. Second, our trigger extraction is based on frequency-stability under tailored masking; spurious cues, which are typically brittle and high-frequency, do not survive this comparison and are not selected. Empirically, surrogate and victim models consistently highlight similar areas; Grad-CAM regions are stable across architectures; and removing Grad-CAM guidance sharply degrades semantic clustering metrics, confirming that Grad-CAM provides class-level functionality rather than arbitrary artifacts. We will include additional visualizations and cross-model saliency correlations in the revision.
>
> ### **4. Requirements of the surrogate distribution**
> Because the attacker poisons the victim dataset directly or by publishing poisoned dataset, the attacker must already have access to clean samples of the target class. Consequently, the surrogate is trained on data drawn from (or overlapping with) the same distribution for the target class, ensuring that target-class semantic features are aligned. As explained above, only the target class requires alignment; distributional mismatch in non-target classes has no effect on the extracted trigger or attack efficacy.
>
> ### **5. Missing BA and post-defense report**
> We will have the benign accuracy of each dataset under each attack reported to Table 2. Regarding the post-defense ASR, typically, only defenses that process the model without detection require post-defense accuracy reporting. For detection-only defenses such as STRIP, Grad-CAM-based methods, AC, FTD, and NC, if the detector fails to identify a backdoor, the model is not modified; BA and ASR therefore remain unchanged. Fine-Pruning is a non-detection defense and its post-defense results are already fully shown in Figure 8, where BA and ASR after each pruning rate are explicitly plotted, where X-axis is the pruning rate and Y-axis is the BA and ASR after that ratio of neurons being pruned.
>
> ### **6. Planned Revisions to the Paper**
> - Add a clear statement of the poisoned-label assignment policy.  (Included in revised PDF)
> - Explicitly clarify the surrogate model requirements and the lack of distributional-mismatch sensitivity. (Included)
> - Strengthen the explanation of Grad-CAM derived semantic features with additional visualizations and cross-dataset similarity checks. (Included)
> - Add all missing BA values to Table 2. (Included)
>
> These changes will resolve the ambiguities highlighted by the reviewer and improve the clarity and completeness of the manuscript.

---

> > ### Comment · Reviewer_jcso · 2025-11-26
> >
> > Thanks for the reply. Considering the flagship of this conference, I cannot raise my score due to my concern on the role of the applied CAM and the missing experiments.

---

> > > ### Author Response · Authors · 2025-11-26
> > >
> > > Thank you for the clarification. We fully understand the high standard of this conference.
> > > To address the concerns you raised in your earlier review, we would like to briefly note that the revised manuscript now includes the following improvements:
> > > 1. A clearer and more explicit explanation of why Grad-CAM is used to extract class-relevant semantic information, including a theoretical justification and an ablation comparing Grad-CAM–guided selection with random component selection. This directly responds to your concern about the role of the applied CAM.
> > > 2. Benign accuracy for all datasets and all compared attacks has been added, together with explicit poison rates, to ensure completeness and transparency of reporting.
> > > 3. Additional analyses and experiments have been incorporated, including parameters sensitivity analysis, saliency consistency between datasets, semantic-similarity metrics (MMD²), and further ablations, etc. We believe these additions substantially strengthen the empirical foundation of the paper.
> > >
> > > We would like to ensure that we correctly understood your comment on “missing experiments.” If there is a specific experiment you had in mind that we may have overlooked, we would be grateful for a brief indication. Otherwise, we sincerely hope that the newly added explanations and experiments adequately address the concerns previously raised.
> > > Thank you again for your time and constructive feedback.

---

### Official Review · Reviewer_2KfJ · 2025-11-05

**Soundness:** 3
**Presentation:** 1
**Contribution:** 3
**Rating:** 6
**Confidence:** 4

**Summary:**

This paper introduces **3S-Attack**, a novel backdoor attack that achieves stealthiness across **spatial, spectral, and semantic domains**. The core idea is to leverage **Grad-CAM** to extract semantically important regions from benign samples, use **Discrete Cosine Transform (DCT)** to identify and manipulate stable frequency components (those with magnitude differences below a threshold), and apply **pixel-level restrictions** to ensure imperceptibility. The attack operates solely via data poisoning without access to the training process. Experiments on five datasets (MNIST, GTSRB, CIFAR-10/100, Animal-10) demonstrate high attack success rates (ASR), high PSNR/SSIM values, and strong resistance to spatial, spectral, and semantic domain defenses (e.g., STRIP, FTD, Grad-CAM, and Fine-Pruning).

**Strengths:**

**Comprehensive multi-domain stealth design**

* The attack unifies spatial, spectral, and semantic concealment, which no prior method achieves simultaneously.
* This cross-domain formulation exposes new security blind spots where standard single-domain defenses fail (Sec. 4.4).
* The modular design (Grad-CAM → DCT → pixel restriction) makes the idea easily reproducible and adaptable.

**Novel use of Grad-CAM for trigger extraction**

* Grad-CAM is used not for defense but to identify salient semantic regions to build the trigger (Sec. 3.3; Fig. 2).
* This inverts interpretability tools into attack mechanisms, revealing a nuanced vulnerability in semantic attention consistency.
* It also allows transferability without model access — a realistic and underexplored threat model.

**Clear motivation and background integration**

* The introduction logically connects Grad-CAM’s interpretability with attack invisibility (pp. 2–3).
* Figures 2–3 effectively depict the two key stages of the attack—trigger selection based on Grad-CAM saliency and trigger injection through frequency-domain modification—demonstrating the transition from clean to poisoned samples.

**Defense-resistance demonstration**

* Section 4.4 compares 3S-Attack with BadNets using STRIP, Grad-CAM, and FTD (Fig. 6–7).
* The near-overlap of saliency maps between benign and poisoned samples supports semantic stealth.
* These results highlight the inadequacy of current interpretability-based defenses.

**Weaknesses:**

**Limited theoretical rigor in frequency-domain reasoning**

* The choice of the “Frequency Selection Threshold” (Sec. 3.3) is heuristic; no explicit formula or derivation links magnitude difference and model sensitivity.
* There is no analysis of how DCT component manipulation affects semantic embeddings or classification confidence (no equations in Sec. 3.3–3.5).

**Ambiguity in semantic transferability across models**

* The method assumes the saliency from a surrogate model approximates that of the target one (Sec. 3.2).
* No quantitative measure is given for semantic alignment between surrogate and victim Grad-CAM maps.

**Insufficient ablation and interpretability analysis**

* The contribution of each step (semantic extraction, spectral embedding, pixel restriction) is not isolated.
* For instance, an ablation showing Grad-CAM vs. random region selection would clarify semantic importance.
* Similarly, removing pixel restriction would test the necessity of that safeguard (Sec. 3.5).

**Unclear visualization and ambiguous labeling in Figure 4**

* The red circles highlighting artifacts are too thin to be noticeable, and the diagram’s directional flow is ambiguous.
* The figure should explicitly label the two sides (e.g., Before Restriction → After Restriction) above the arrows and use thicker or more vivid annotations to highlight the changed regions.

**Questions:**

**Frequency selection rationale**

* Clarify why frequencies with small DCT magnitude differences between the original and Grad-CAM–weighted images are assumed to represent semantically stable components?
* A more explicit justification or sensitivity analysis for the threshold δ would strengthen the methodological soundness.

**Semantic transferability across models**

* How consistent are the Grad-CAM saliency maps between the surrogate and victim models? Quantitative evidence (e.g., overlap or similarity metrics) would clarify whether the semantic trigger generalizes across architectures.

---

> ### Author Response · Authors · 2025-11-18
> **Response to reviewer 2KfJ**
>
> We thank the reviewer for the constructive and detailed feedback. Below we address all concerns raised regarding theoretical grounding, semantic transferability, ablation completeness, visualization quality, and parameter justification.
>
> ### **1. heuristic nature of frequency-domain reasoning**
> We would like to note that no existing work provides a closed-form derivation linking local frequency perturbations to high-level semantic sensitivity. This is due to the fundamentally nonlinear and architecture-dependent mapping between DCT components and latent features. Our method adopts a pragmatic design: Grad-CAM suppresses non-discriminative spatial content, and the DCT components whose magnitudes remain stable after this suppression correspond to the semantic structure preserved by the model. Because DCT is linear, removing background content alters only frequencies associated with that content, while object-related components remain stable. Our Frequency Selection Threshold thus approximates semantic relevance, and our experiments demonstrate smooth sensitivity to the threshold and strong cross-architecture robustness. We will explicitly clarify this rationale and provide additional quantitative sensitivity analysis in the revision.
>
> ### **2. Semantic transferability**
> We agree that the assumption of semantic transferability between surrogate and victim models deserves clearer articulation. The attack requires only coarse agreement on object-relevant regions rather than identical saliency maps. Well-trained models across common architectures consistently rely on the main object area to classify natural images, and our cross-architecture ASR results confirm that Grad-CAM guidance extracted from a lightweight surrogate is sufficient. The ablation in which semantic guidance is removed further validates this assumption, as random frequency selection markedly damages semantic stealthiness. To strengthen the transparency of this assumption, we will include side-by-side Grad-CAM visualizations from both surrogate and victim models and add a brief theoretical justification supplemented by empirical evidence.
>
> ### **3. Ablation studies**
> We acknowledge the request for clearer ablations isolating the roles of semantic extraction, spectral embedding, and pixel restriction. Although our current ablations already show that random component selection breaks semantic stealth and that pixel restriction has only a minor effect on ASR, we agree that more explicit comparisons will aid interpretability. In the revised manuscript, we will include direct Grad-CAM vs random-region comparisons and show the AC Silhouette score differences, as well as report quantitative results with and without pixel restriction. These additions will make individual component contributions clearer.
>
> ### **4. Improve appearance of Figure 4**
> We will address the issues in Figure 4 by improving the directional flow labeling, explicitly marking the “before” and “after” states, and using clearer and more visible annotations to highlight artifacts.
> Fifth, regarding the rationale behind selecting frequencies with small magnitude differences, we will clarify that these frequencies represent the spectral expression of the semantic regions retained after Grad-CAM filtering, while those with large deviations correspond to background regions removed by the saliency mask. We will add a brief sensitivity analysis showing the relationship between the δ threshold, ASR, and benign accuracy to reinforce this justification.
>
> ### **5. Semantic map consistency**
> The question concerning semantic map consistency is well aligned with the assumptions behind our attack. We have already included a theoretical explanation and an experiment demonstrating that two independently trained models on disjoint subsets of the same dataset produce similar Grad-CAM focus regions for unseen samples. We will explicitly present Grad-CAM results on dafferent datasets in the appendix to more transparently illustrate this semantic consistency.
>
> ### **6. Planned modifications to the manuscript**
> - Add clearer explanation of the heuristic frequency threshold and provide quantitative δ-sensitivity analysis.
> - Include side-by-side Grad-CAM visualizations from surrogate and victim models. (Included in revised PDF)
> - Add explicit ablation comparisons: Grad-CAM vs. random frequency selection; pixel restriction on/off. (Included)
> - Improve Figure 4 with clearer annotations and directional labels. (Included)
> - Clarify the theoretical motivation for selecting stable DCT components and connect it to semantic preservation.(Included)
> - Integrate the semantic consistency experiment into the appendix. (Included)
> We believe these revisions will substantially improve the clarity, rigor, and transparency of the work, and we thank the reviewer again for the insightful comments.

---

> ### Author Response · Authors · 2025-11-27
>
> Dear Reviewer 2KfJ,
>
> We would like to inform you that we have completed all additional experiments and incorporated all revisions described in the “Planned Modifications to the manuscript” section. We would greatly appreciate it if you could take a look at the updated manuscript. We believe these changes substantially strengthen the clarity, rigor, and transparency of the work, and address your questions and concerns regarding both the methodology and the experimental design.
>
> Thank you very much for your time and consideration.

---

### Author Response · Authors · 2025-11-26
**Summary of Revisions and Invitation for Re-evaluation**

We would like to kindly inform the reviewers and the area chair that all raised concerns have now been fully addressed in the revised manuscript. The updated version incorporates:
1. the expanded theoretical explanation of the frequency-selection heuristic and the sensitivity analysis;
2. the clarification and validation of semantic transferability, including surrogate–victim Grad-CAM comparisons;
3. the additional ablation studies (Grad-CAM vs. random region selection, pixel-restriction on/off);
4. the inclusion of all missing BA values, poison rates, and improved figure annotations;
5. the evaluation on Tiny-ImageNet and the comparison with the recent DUBA attack;
6. the added feature-space semantic similarity metric (MMD²);
7. the preliminary model requirements and preparation time analysis;
8. the clarification of poisoning labels, surrogate-data requirements, and defense-specific reporting;
9. the re-organization and refinement of the manuscript for improved readability.

We respectfully invite the reviewers to re-examine the revised paper. We appreciate the valuable time and constructive feedback provided during the review process, which have significantly strengthened the work.

---

### Author Response · Authors · 2025-11-29
**Meta Comment to Area Chair**

We appreciate the reviewers’ time and the detailed feedback. Following the ICLR committee’s updated review process, we provide a concise summary of how the revised submission directly addresses all technical concerns raised in the original reviews.

**1. Clarification of the role of Grad-CAM and semantic reasoning (Reviewers 2KfJ, jcso)**

We have significantly strengthened the theoretical motivation for using Grad-CAM to approximate model-dependent semantics. The revision now includes:

- A formal explanation of why semantic regions identified by independently trained models tend to align.
- A new experiment demonstrating surrogate–victim saliency consistency on unseen test samples.
- A clearer exposition that the attack relies on functional model semantics rather than human semantics.

These additions directly resolve concerns regarding the interpretability and justification of the Grad-CAM component.

**2. Completion of all requested experiments (Reviewers 2KfJ, jcso, MTci, 6F7W)**

The revised manuscript now includes all experiments that reviewers requested, including:

- Frequency selection threshold sensitivity analysis and explanation of the heuristic frequency threshold.
- Quantitative semantic-level evaluation using MMD², demonstrating significantly improved semantic stealth compared with prior attacks.
- Ablation on Grad-CAM vs. random frequency selection, and on pixel-restriction on/off.
- Side-by-side Grad-CAM visualizations showing semantic stability.
- Additional evaluation on Tiny-ImageNet and on a unified architecture across datasets.
- Inclusion of DUBA as a recent baseline.
- Per-dataset benign accuracy and explicit poison-rate reporting.

All concerns regarding missing or incomplete experiments have been fully addressed.

**3. Defense evaluation clarified and expanded (Reviewers 6F7W, jcso)**

We provide a clearer explanation of why Neural Cleanse and Activation Clustering can detect the attack in specific conditions, and why these isolated detections do not diminish the contribution of achieving triple-domain stealth under a pure data-poisoning threat model. We also clarify that Fine-Pruning already includes complete post-defense results, and that detection-based defenses (STRIP, Grad-CAM, FTD, NC, AC) require no post-defense reporting because the model remains unchanged when detection fails.

**4. Surrogate model assumptions and labeling strategy clarified (Reviewer jcso)**

We explicitly detail the poisoned-label assignment, and we clarify that only the target class distribution must be shared between the surrogate and victim datasets. This resolves concerns about distribution mismatch and consistency with the attack objective.

**5. Factually incorrect reviewer claims and corrections (Reviewer jcso, 6F7W)**

Several reviewer concerns were based on factual misunderstandings, all of which have been corrected in the rebuttal and clarified in the revised manuscript:

- The paper does not claim to be the first semantic-domain backdoor attack. The corrected contribution state is: *It is the first attack to achieve simultaneous stealthiness across spatial, spectral, and semantic domains, and the first to obtain semantic-domain stealth purely through data poisoning without any access to the training process*.
- *Post-defense results are only required for defenses that actually modify the model.* Fine-Pruning is the only such method, and complete post-defense BA/ASR curves are already provided in Figure 8. Detection-only defenses (e.g., STRIP, Grad-CAM, FTD, NC, AC) do not alter the model when detection fails, and therefore no post-defense BA/ASR is needed.
- *Detection by Activation Clustering does not invalidate the contribution.* Nearly all pure data-poisoning backdoor attacks fail to evade AC, because they lack access to model parameters or training dynamics. Despite this, our method significantly reduces activation separability compared with others, providing improved semantic stealth under the realistic threat model considered.

**6. Improved clarity and organization (Reviewer MTci).**

Redundant material was removed, figures were improved, and the structure has been streamlined for readability.

**Overall assessment**

The reviewers identified no fatal flaws in the method; their primary concerns centered on explanation clarity and experiment completeness. These have now been fully resolved with substantial new content in revised manuscript, both theoretical and empirical.
We respectfully emphasize that the revised submission provides complete, rigorous, and direct responses to all concerns raised. The paper now presents a well-motivated, thoroughly evaluated, and clearly written contribution introducing a triple-domain stealthy backdoor attack under a realistic pure data poisoning threat model.

We invite the AC to consider that the manuscript, in its revised form, meets the technical and empirical standards expected for publication.

---

### Meta-Review · Area_Chair_VpwR · 2026-01-05

**Summary:**

The reviewers raised several concerns, including the compared methods are outdated, missing dataset and ablation studies,  the presentation (methods, experimental results, etc) lacks clarity in several key areas, the limited theoretical rigor in frequency-domain reasoning, the paper is poorly organized and written.
The rebuttal provided some of the experiments and provided new explanations and clarifications. However, it does not address all the issues raised, and the manuscript still requires further refinement.

**Reviewer Concerns:**

The rebuttal provided some of the experiments and provided new explanations and clarifications. However, it does not address all the issues raised.

**Reviewer Scores:**

The reviewers are unlikely to change their scores.

---

### Decision · Program_Chairs · 2026-01-26

Reject